# Breath Acetone Measurement-Based Prediction of Exercise-Induced Energy and Substrate Expenditure

**DOI:** 10.3390/s20236878

**Published:** 2020-12-01

**Authors:** Min Jae Kim, Sung Hyun Hong, Wonhee Cho, Dong-Hyuk Park, Eun-Byeol Lee, Yoonkyung Song, Yong-Sahm Choe, Jun Ho Lee, Yeonji Jang, Wooyoung Lee, Justin Y. Jeon

**Affiliations:** 1Department of Sport Industry Studies, Yonsei University, Seoul 03722, Korea; spopd@naver.com (M.J.K.); hyun-ny-love@hanmail.net (S.H.H.); zznh1129@naver.com (W.C.); wosan2@naver.com (D.-H.P.); qufdl830@naver.com (E.-B.L.); lapine1019@yonsei.ac.kr (Y.S.); 2Isenlab Inc., Gyeonggi-do 13215, Korea; cys@isenlab.com; 3Department of Materials Science and Engineering, Yonsei University, Seoul 03722, Korea; jayhnj1221@naver.com (J.H.L.); yeonzi-_-@naver.com (Y.J.)

**Keywords:** energy metabolism, substrate oxidation, lipid oxidation, ketone bodies, breath acetone, breath acetone analyzer, excess post-exercise oxygen consumption

## Abstract

The purpose of our study was to validate a newly developed breath acetone (BrAce) analyzer, and to explore if BrAce could predict aerobic exercise-related substrate use. Six healthy men ran on a treadmill at 70% of maximal oxygen consumption (VO_2max_) for 1 h after two days of a low-carbohydrate diet. BrAce and blood ketone (acetoacetate (ACAC), beta-hydroxybutyrate (BOHB)) levels were measured at baseline and at different time points of post-exercise. BrAce values were validated against blood ketones and respiratory exchange ratio (RER). Our results showed that BrAce was moderately correlated with BOHB (*r* = 0.68, *p* < 0.01), ACAC (*r* = 0.37, *p* < 0.01) and blood ketone (*r* = 0.60, *p* < 0.01), suggesting that BrAce reflect blood ketone levels, which increase when fat is oxidized. Furthermore, BrAce also negatively correlated with RER (*r* = 0.67, *p* < 0.01). In our multiple regression analyses, we found that when BMI and VO_2max_ were added to the prediction model in addition to BrAce, R^2^ values increased up to 0.972 at rest and 0.917 at 1 h after exercise. In conclusion, BrAce level measurements of our BrAce analyzer reflect blood ketone levels and the device could potentially predict fat oxidation.

## 1. Introduction

The continuous rise of obesity prevalence is a global trend [1]. Obesity causes multiple health problems, such as type 2 diabetes, cardiovascular disease, hypertension, dyslipidemia, stroke, musculoskeletal disorders, dementia, depression, and cancer [2,3,4]. Moreover, obesity results in significant medical expenses, which rise continually given the increasing number of people with obesity [5,6].

To prevent chronic diseases caused by obesity, the World Health Organization (WHO) recommends reducing caloric intake and increasing energy expenditure through exercise. Physical activity (PA) coupled with a proper diet is an effective weight management approach without side effects [7,8]. Total energy expenditure (EE) consists of three domains: resting metabolic rate (RMR), PA-associated energy expenditure, and the thermic effect of food (TEF). RMR and TEF generally account for approximately 60–75% and 10% of 24 h EE, respectively. Exercise increases energy cost as well as RMR when muscle mass increases as a result [7,9].

Exercise intensity and duration are associated with energy and substrate expenditure based on an individual’s diet and training status [10]. In order to better assess the level of energy and substrate expenditure during exercise, a metabolic cart has been extensively used, which calculates energy and substrate expenditure based on oxygen consumption (VO_2_) and respiratory exchange ratio (RER). However, this device can only be used in the laboratory and also very expensive. Recently, wearable devices provide information on energy expenditure based on heart rates and accelerometry data [11,12], but are not able to provide substrate utilization during exercise. In this regard, breath acetone became a good candidate to be used to estimate the substrate utilization during exercise or PA [13,14].

As fat oxidation increases, the level of ketone bodies, in the form of acetoacetate (ACAC), beta-hydroxybutyrate (BOHB), and breath acetone (BrAce), also increases [15]. All ketone bodies are present in the blood and circulate through the bloodstream. Due to the volatile nature of acetone, it fills the lungs and is exhaled with the breath [16,17]. A few studies have reported a moderate association between BrAce and blood ketone levels [18] either at rest [19] and during ketogenic diet [20]. There was one study which examined BrAce as well as BOHB in response to 90 min interval exercise on bicycle. Although Guntner et al. reported that increase in BrAce levels may have reflected increase in BOHB, which may suggest increased fat oxidation, they did not directly measure fat oxidation using a metabolic cart, nor measured free fatty acid (FFA) in the blood [21]. Furthermore, they only collected blood three times throughout the course of study including at baseline, immediately post exercise and three hours after exercise cessation. Previous studies only examined correlation between BrAce and blood BOHB, yet did not examined whether BrAce would associate with metabolic cart measured fat utilization or plasma FFA concentration. Therefore, it is necessary to study whether BrAce would be associated with other blood ketone such as ACAC or BOHB, and more importantly with plasma FFA and metabolic cart accessed fat utilization.

Recently, we developed a portable-size breath analyzer using a SnO_2_ nanorod (NR)-based sensor with a miniaturized Gas Chromatography (GC) column, highly sensitive (detection limit: 0.1 ppm), ideal for human breath acetone analysis [22]. Therefore, our primary purpose was to validate our newly developed BrAce analyzer at rest, during one hour of moderate-intensity exercise, and during three hours of recovery, whether BrAce would associate with blood ketones including ACAC and BOHB. Then, we further explored if BrAce could be an indicator of substrate utilization assessed by a metabolic cart as well as plasma FFA before, during, and after aerobic exercise.

## 2. Materials and Methods

### 2.1. Participants

A total of six healthy men participated in this study. The inclusion criteria were as follows: (1) men aged between 18–45 years, (2) no physical problems determined by the Physical Activity Readiness Questionnaire (PAR-Q) [23], (3) moderately trained runners (≥5 km/week), and (4) could perform running at 70% of maximal VO_2_ (VO_2max_) for more than 60 min. The experimental procedure was fully explained to all participants by the experimenter, and all participants gave written informed consent prior to the test. This study was approved by the Institutional Review Board of Yonsei University (Yonsei IRB no. 7001988-201910-HR-674-05).

### 2.2. Experimental Protocol

Participants visited the laboratory twice for the study. They were asked to refrain from smoking and drinking alcohol for at least 24 h prior to the measurements. During their first visit, PAR-Q survey, anthropometric measurements, and VO_2max_ were measured. As the amount of glycogen stored in the liver and muscles might influence substrate utilization, the participants were asked to consume less than 80 g of carbohydrate (CHO) per day for two days [24]. After checking their food journal to make sure that they consumed less than 80 g of CHO per day for two days prior to the second visit, the participants ran on a treadmill for one hour at a moderate intensity (70% VO_2max_). During the test, the VO_2_ and RER were continuously monitored using a metabolic cart (Cardiac Science Co., Waukesha, WI, USA). Energy expenditures from fat and CHO oxidation were calculated according to the non-protein respiratory quotient based on the VO_2_ and RER obtained from the metabolic cart [25]. BrAce values were measured 6 times in total: at the baseline, immediately after exercise, as well as after 30, 60, 120, and 180 min of recovery.

### 2.3. Measurements

#### 2.3.1. Anthropometric Measurements

Body weight and height were measured using an electric extensometer (BDM 330, Biospace, Seoul, South Korea) to the nearest 0.1 kg and 0.1 cm, respectively. The body mass index (BMI) was described as kg/m^2^. The participants were asked to wear lightweight clothing with no shoes. Body composition was assessed using bioelectrical impedance analysis (Inbody 720, Biospace, Seoul, South Korea) [26,27,28].

#### 2.3.2. Cardiopulmonary Exercise Test

The cardiopulmonary exercise test was performed on a treadmill using a computerized cardiac stress testing system (Cardiac Science, Q-stress TM65, Waukesha, WI, USA) to measure VO_2max_ which would be used as a reference of exercise intensity. The participants wore a non-breathing facemask (Hans Rudolph, Rudolph series 7910, Kansas, MO, USA) during the test. The volume of carbon dioxide in exhaled air was continuously analyzed for single-breath measurement using a computerized metabolic measurement system (ParvoMedics, TrueOne 2400, Salt Lake City, UT, USA). A well-trained investigator followed the Bruce protocol [29,30]. VO_2_ was considered maximal if any two of the following three criteria were met: (1) RER of > 1.15; (2) heart rate of > 85% of age-predicted maximal heart rate; and (3) perceived exertion rate of ≥ 17 on the Borg scale. Measured VO_2max_ was used to establish the relative exercise intensity according to the participants.

#### 2.3.3. BrAce Analyzer

Only end-tidal breath was collected in 50 mL sample bag. Then, this bag was connected to the BrAce analyzer. The BrAce analyzer was composed of a sampling loop, a packed column, three solenoid valves, a mini-sized pump, and a SnO_2_ NR-based sensor (Figure 1). The detailed explanation of the sensors, characterization and procedure of the working valve and pump have been previously described [22,31]. The size of the BrAce analyzer was 8 × 13 × 16 cm^3^_,_ which is portable. Once the sampling loop was filled with 1 mL of exhaled breath, it passed through the packed column within 100 s and was separated by the difference in interaction with the stationary phase. The stationary phase the property of retaining polar molecules rather than non-polar molecules, separation occurred due to the difference in polarity and mass of the molecules. Acetone has polar property, separated from the column and detected in 30 s. The separated acetone was detected using the SnO_2_ NR-based sensor. The SnO_2_ NRs were fabricated by the glancing angle deposition (GLAD) method via e-beam evaporation. We used commercial SnO_2_ granules (4N purity, Kojundo Chemical Laboratory Co., Ltd., Sakado-shi, Japan) as SnO_2_ sources for fabrication. BrAce analyzer used in our experiment was validated with GC-Mass Spectrometry (MS) and prior to each test, the device was calibrated ranged from 0.1–50 ppm of acetone vapor [22].

#### 2.3.4. Blood Sampling

Blood samples were taken six times: at baseline, immediately after 1 h of exercise, and 30, 60, 120 and 180 min after exercise. Blood samples (10 mL each time) were collected by standard vein puncture into the plain tube. Then blood samples were centrifuged at 3000 rpm for 10 min. The serum was separated into a separate tube and kept in −80 °C freezer until further analysis of the following variables: glucose, BOHB, ACAC, blood ketone (BOHB + ACAC), and FFA.

#### 2.3.5. Statistical Analysis

All data were analyzed using the statistical package SPSS version 25.0 (IBM, Armonk, NY, USA). The results were expressed as mean ± standard error, and statistical significance was set at *p* < 0.05. The correlation analysis was conducted on BrAce, blood ketone, FFA, and RER. To calculate the estimated equation for deriving lipid oxidation using BrAce, we performed multiple regression analyses with the related variables.

## 3. Results

### 3.1. Characteristics of the Participants

The characteristics of the participants are shown in Table 1. The mean age of the participants was 27.0 ± 1.8 years and their VO_2max_ was 51.7 ± 3.7 mL/kg/min. None of our participant was a smoker. The level of BrAce prior to the exercise was 2.9 ± 0.8 ppm.

### 3.2. Energy Expenditure and Substrate Oxidation at Rest, During and after Exercise

Table 2 illustrates energy expenditure and contributions of CHO and fat oxidation to EE during rest, exercise, and post-exercise for 3 h. Prior to exercise, the rate of fat oxidation was significantly higher than that of CHO (64.2 ± 5.2% vs. 35.8 ± 5.2%). Participants used a total of 751.6 ± 45.8 kcal for 1 h of running exercise. The energy used during exercise was provided from 52.8 ± 4.7% of CHO and 47.2 ± 4.7% of fat. During the 3 h of recovery, the participants used a total of 309.2 ± 24.9 kcal. The rate of fat oxidation was significantly higher than that of CHO oxidation throughout the 3 h post-exercise period (Post-Ex 1 h: 23.6 ± 6.6 kcal/h vs. 82.7 ± 7.0 kcal/h, Post-Ex 2 h: 23.8 ± 6.4 kcal/h vs. 80.0 ± 5.4 kcal/h, Post-Ex 3 h: 28.2 ± 9.6 kcal/h vs. 70.9 ± 6.7 kcal/h for CHO vs. fat oxidation, respectively).

### 3.3. BrAce and Blood Ketone at Rest, during, and after Exercise

Since fat oxidation was the highest at 1 and 2 h of post-exercise recovery, then started to decrease after 3 h of post-exercise recovery, we expected that BrAce would increase until Post-Ex 2 h, then start to decrease during Post-Ex 3 h. However, BrAce steadily increased over the 3-h post-exercise period (Figure 2a). The blood ketone (BOHB and ACAC) levels showed similar responses to BrAce, which continuously increased until 120 min post-exercise (Figure 2b,c). Notably, there was a decrease in the blood ketone concentration after Post-Ex 2 h. FFA levels increased during the exercise. However, they started to decrease immediately after cessation of the exercise and remained higher than their level during the rest period (Figure 2d).

### 3.4. Correlation between BrAce, Blood Ketone, and RER

First, we examined whether BrAce reflects blood ketone levels. BrAce levels significantly correlated both with BOHB (*r* = 0.68, *p* < 0.001) (Figure 3a) and ACAC (*r* = 0.37, *p* = 0.028). Then, we further examined whether BrAce, as well as blood ketone levels, could be associated with RER, which reflects substrate utilization. RER significantly correlated with BrAce levels (*r* = −0.67, *p* < 0.001), BOHB (*r* = −0.71, *p* < 0.001), and ACAC (*r* = −0.58, *p* < 0.001) (Figure 3b–d). We examined whether these correlations might vary according to the different stages of testing (rest, exercise, and post-exercise) and found moderate to strong correlations between the variables regardless of the stage of testing.

### 3.5. Predictability of Lipid Oxidation Using Breath Acetone

Since we confirmed that BrAce levels reflect blood ketone levels, as well as RER, we developed a model to predict the amount of fat oxidation using BrAce. As shown in Table 3, BrAce, BMI, and VO_2max_ were used to predict fat oxidation at rest and during recovery from exercise. At rest, BrAce was used to predict fat oxidation, with an adjusted R^2^ value of 0.667 (Model 1). When the VO_2max_ value was added to the model (Model 3), the R^2^ values increased to 0.97. Similarly, when BrAce at 1 h of recovery and BMI were used to predict fat oxidation during recovery, the adjusted R^2^ value was 0.546 (Model 2), but when VO_2max_ value was added to the model (Model 3), the adjusted R^2^ value increased to 0.917 (Model 3). The BrAce value at 2 and 3 h of recovery did not predict the fat oxidation level during recovery, we thus developed a prediction equation for fat oxidation for the rest and 1 h post-exercise periods.

## 4. Discussion

The purpose of this study was to validate whether BrAce measured at rest and during exercise recovery would reflect blood ketone levels and the level of fat oxidation (Figure 4). As hypothesized, we observed a significant association between BrAce and blood ketone levels, as well as other markers for fat oxidation, such as RER and plasma FFA levels. Our study is unique in that BrAce was validated not only during at rest but after 1 h of moderate-to-high intensity treadmill running, which resulted in significant physiological changes. Other studies reported strong correlations between BrAce and BOHB, with an average value of *r* = 0.77 (range: 0.54–0.94) [32,33,34,35,36,37]. In our study, we found moderate to strong correlations among BrAce, blood ketone, and RER (Figure 3) levels. These results were comparable to those of other studies that found *r* value between BrAce and BOHB of 0.68 (*p* < 0.001) [32,34,37]. The enzymatically controlled metabolic pathway could produce different linear relationships between blood ketone and BrAce levels. One of the important findings in our study was the moderate to strong correlations between BrAce and RER (*r* = −0.67, *p* < 0.001), indicating that BrAce highly correlated with fat oxidation. Moreover, we analyzed correlations among variables at each phase: rest, during the exercise, and recovery periods. We found moderate to strong associations between the variables at each phase, suggesting that BrAce could be a potential biomarker of fat oxidation both at rest and after exercise.

In our study, we found that fat oxidation was 1.79 times more than CHO oxidation (64.2% vs. 35.8%) at rest, while the ratio of CHO oxidation was higher than that of fat oxidation during the 1-h moderate-to-high intensity exercise (52.8% vs. 47.2%) (Table 2). In the post-exercise period, the fat oxidation rate was 2.88 to 3.90 times higher than the ratio of CHO oxidation (Post-Ex 1 h: 79.6% vs. 20.4%, Post-Ex 2 h: 79.2% vs. 20.8%, and Post-Ex 3 h: 74.2% vs. 25.8% for lipid and CHO, respectively). Consuming a low-CHO and high-fat diet could cause the body to shift from using CHO to fat as its primary fuel. Therefore, it is considered that carrying out 48 h of ketogenic diet led our participants to prioritize using fat over CHO during the rest period. However, the opposite results were found during the 1-h moderate-to-high intensity treadmill running that CHO was more likely to be used as a substrate than fat. This is supported by previous studies showing that the type of energy source is determined by the intensity of exercise, and the rate of CHO metabolism increases with increasing exercise intensity [38,39,40]. The EE increased during the exercise and decreased significantly after the cessation of the exercise. Although EE decreased after the exercise, its absolute value was higher than that during the resting period in order to compensate for the energy deficit [41]. However, the fat oxidation rate was higher than that of the CHO oxidation during all periods, except for the period of the exercise [42].

Although multiple studies have shown the effect of the exercise on fat oxidation (i.e., BOHB), the effect of exercise on BrAce has not yet been clearly investigated. Guntner et al. examined BrAce and blood BOHB in response to 90 min of interval exercise on bicycle. They reported increased BrAce and BOHB during 3 h of resting after exercise [21]. Similarly, Bovey et al. examined whether a 2-h low-intensity exercise would increase BrAce levels during and after exercise at three different energy balance statuses (fasting, low CHO, and high CHO) [43]. They found a significant increase in BrAce and BOHB only when the participants exercised while fasting and suggested a negative energy balance due to exercise-induced ketone production. This result is similar to ours in that all our participants fasted during exercise and recovery, which resulted in a significantly negative energy balance. In our study, BOHB did not increase significantly during exercise. However, it significantly increased immediately after exercise and during recovery. Nevertheless, plasma FFA increased during exercise but did not further increase during recovery. The lack of significant changes in BOHB and BrAce during exercise could be explained by the exercise intensity applied in our study. Moderate-to-high intensity exercise restricted blood flow to the liver, and as a result, blood ketone production could have been reduced during high-intensity exercise in our study. Then, ketone production significantly increased during recovery, probably due to an increase in the blood flow to the liver and a negative energy balance status resulting from exercise.

Another interesting finding is that BOHB and ACAC started to decrease after 2 h of exercise recovery, while BrAce continued to increase up to 3 h of recovery after exercise. This result could be explained by the fact that BrAce and BOHB differentiate not only to changes in ACAC production rate but also to relative changes of ACAC conversion into BrAce and BOHB [44]. Another natural explanation for this phenomenon could be the necessary time delay for BrAce to reflect blood ketone levels. However, a clear understanding of this phenomenon is beyond the scope of our study.

To test whether BrAce could be used to predict fat oxidation at rest, during exercise, and exercise recovery, we conducted multiple regression analyses using BMI and fitness levels of the participants. When BMI of the participants was added to the model in addition to BrAce, it did not increase the R^2^ value. However, when VO_2max_ was further added to the model, we observed a significant increase in the R^2^ value, suggesting that the aerobic fitness level of the participants, in addition to BrAce levels, increased the estimation of fat oxidation. Similar patterns were observed when developing equation models to predict fat oxidation using BrAce at rest and after exercise. Based on these multiple regression analyses, BrAce levels at 1 h post-exercise would be ideal for predicting fat oxidation during exercise recovery. However, due to the complexity and cost of the VO_2max_ measurements, this model should be modified for general population use. Instead of the VO_2max_ measurements, either heart rate response during exercise [45,46] or resting heart rate could be used to estimate VO_2max_ [47]. Previously, Guntner et al. also reported significant increase in BrAce and blood BOHB after exercise, which they claimed that increased BOHB was marker for fat oxidation [21]. The difference between our study and the study of Guntner is the frequency of sampling, exercise protocol, and index of fat oxidation. We continuously assessed substrate utilization with indirect calorimetry (metabolic cart) throughout the study period and further measured plasma FFA six times including immediately after the exercise, post-exercise of 30, 60, 120 and 180 min. Therefore, our study provided significantly more information on substrate utilization including fat oxidation, which we compared with BrAce levels. However, due to small sample size of our study, we cannot say with confidence that the model developed in our study could be applied to general population, rather, as we clearly mentioned in the purpose of the study, we explore the possibility of using BrAce to predict substrate utilization. Indeed, we believe that BrAce could be used to predict fat oxidation for many different purposes such as monitoring effect of low CHO diet and exercise.

Like all studies, ours also has several limitations. First, the characteristics of the participants in our study were limited to Asian men with high aerobic fitness levels. Second, to maximize ketosis and fat oxidation, all participants underwent two days of caloric restriction, consuming a low-CHO and high-fat diet. Therefore, our predicted equations may not properly estimate those who consume a high-CHO diet. Last but not least, the number of participants was too small to verify the model developed in our study. Therefore, a study with a larger sample size, including women and participants with lower fitness levels, should be conducted to develop a more accurate and generalizable prediction equation. However, despite these limitations, we explored the possibility of using BrAce for real-time fat oxidation level measurements using a portable-sized, self-monitoring device. Therefore, it could be a useful tool for people who want to set exercise and diet goals for obesity management in the future.

## 5. Conclusions

Six active men were included in this study to investigate substrate oxidation upon a 1-h moderate-intensity treadmill running, and to identify correlations among BrAce, blood ketone, and RER. Additionally, we explored lipid oxidation predictability using BrAce. We found differences between the lipid oxidation ratio and absolute quantity during resting, exercise, and post-exercise period. This study also provided evidence of the significant correlations between BrAce, blood ketone, and RER. In addition, we described a non-invasive, real-time lipid oxidation monitoring approach using a BrAce analyzer.

## Figures and Tables

**Figure 1 sensors-20-06878-f001:**
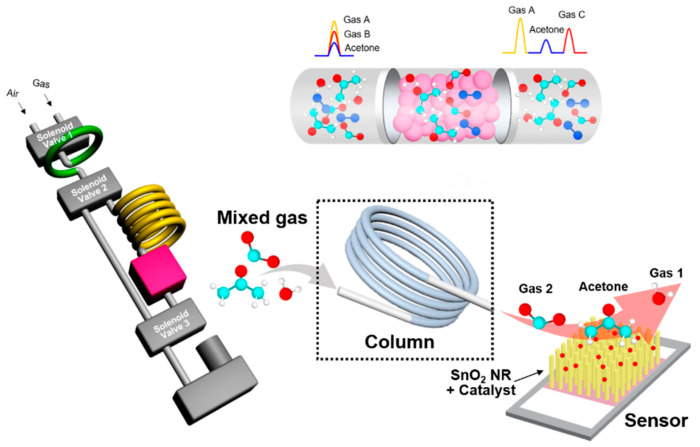
Schematic representation and real image of the breath acetone analyzer.

**Figure 2 sensors-20-06878-f002:**
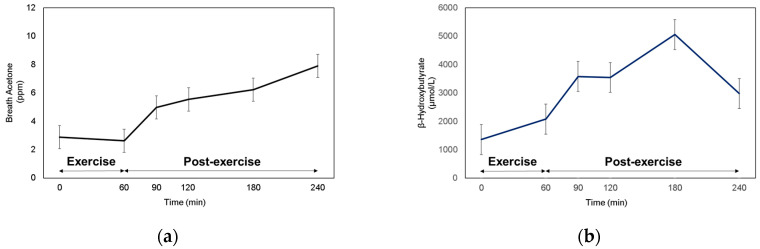
Monitoring of (**a**) Breath acetone, (**b**) β-hydroxybutyrate, (**c**) Acetoacetate, and (**d**) Free fatty acid.

**Figure 3 sensors-20-06878-f003:**
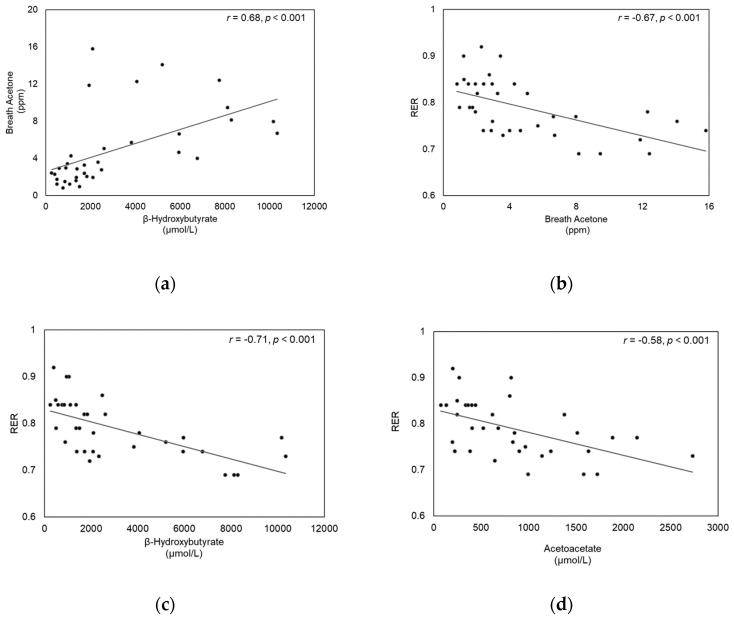
Correlation between (**a**) Breath acetone and β-hydroxybutyrate, (**b**) RER and Breath acetone, (**c**) RER and β-hydroxybutyrate, and (**d**) RER and Acetoacetate.

**Figure 4 sensors-20-06878-f004:**
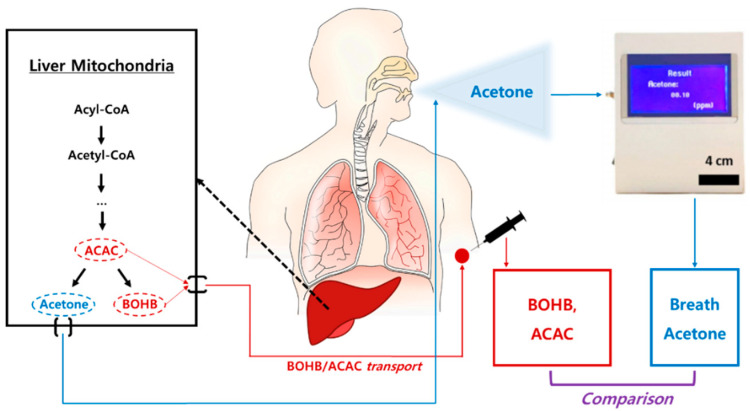
The flow of ketone bodies and their measurements.

**Table 1 sensors-20-06878-t001:** Characteristics of the participants.

Variables	Total (*n* = 6)
Age (years)	27.0 ± 1.8
Height (cm)	178.9 ± 5.3
Weight (kg)	79.6 ± 8.8
Body Mass Index (kg/m^2^)	24.4 ± 1.3
Skeletal Muscle Mass (kg)	39.2 ± 4.5
Body Fat Mass (kg)	11.6 ± 1.5
Body Fat Percent (%)	14.5 ± 0.9
VO_2max_ (mL/kg/min)	51.7 ± 3.7

Data are presented as mean ± standard error. *n*, number; VO_2max_, maximal oxygen consumption.

**Table 2 sensors-20-06878-t002:** Energy expenditure and substrate oxidation at rest, during, and after exercise.

Period	Total	Carbohydrate		Lipid	
	kcal/h	kcal/h	%	kcal/h	%
Resting	76.3 ± 5.4	26.3 ± 6.3	35.8 ± 5.2	50.0 ± 5.2	64.2 ± 5.2
Exercise (1 h)	751.6 ± 45.8	419.8 ± 53.2	52.8 ± 4.7	331.8 ± 30.5	47.2 ± 4.7
Post-Ex 1 h	106.3 ± 7.6	23.6 ± 6.6	20.4 ± 5.2	82.7 ± 7.0	79.6 ± 5.2
Post-Ex 2 h	103.8 ± 7.9	23.8 ± 6.4	20.8 ± 4.6	80.0 ± 5.4	79.2 ± 4.6
Post-Ex 3 h	99.1 ± 9.9	28.2 ± 9.6	25.8 ± 6.8	70.9 ± 6.7	74.2 ± 6.8

Data are described as mean ± standard error. *p* < 0.05, significantly different from carbohydrate. Post-Ex 1 h, 2 h, and 3 h indicate the 1st, 2nd, and 3rd hours of post-exercise period, respectively.

**Table 3 sensors-20-06878-t003:** Prediction equation model development of lipid oxidation using breath acetone.

		Adjusted R^2^	SEE	ΔF	Intercept	BrAce	BMI	VO_2max_
At rest	Model 1	0.667	7.376	11.015 *	38.498 *	2.576 *		
	Model 2	0.584	8.251	4.503	25.667	2.640	0.514	
	Model 3	0.972	2.143	58.677 *	−331.492 *	3.800 *	8.689 *	2.951 *
Post-Ex 1 h	Model 1	0.486	10.619	5.736	60.247 *	2.598		
	Model 2	0.546	9.986	4.005	104.339	2.317	−1.743	
	Model 3	0.917	4.260	19.502 *	−241.492	2.945 *	6.183	2.883
Post-Ex 2 h	Model 1	0.276	12.770	2.904	58.323 *	2.163		
	Model 2	0.054	14.598	1.141	43.198	2.363	0.569	
	Model 3	0.284	12.697	1.661	−449.706	4.064	12.129	3.876
Post-Ex 3 h	Model 1	0.275	19.113	2.899	39.971	2.686		
	Model 2	0.340	18.243	2.287	−55.024	3.804	3.509	
	Model 3	0.070	21.655	1.125	−223.727	4.066	7.428	1.372
Equations								
Fat oxidation at rest			y = −331.492 + 3.800X_1_ + 8.689X_2_ + 2.951X_3_					
Fat oxidation during 3 h of recovery			y = −241.492 + 2.945X_1_ + 6.183X_2_ + 2.883X_3_					

SEE, standard error of estimate; BMI, body mass index; VO_2max_, maximal oxygen consumption; X_1_, BrAce; X_2_, BMI; X_3_, VO_2max_. * *p* < 0.05.

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
