# Peer review of "Breath Acetone Measurement-Based Prediction of Exercise-Induced Energy and Substrate Expenditure"

_sensors, 2020, doi:10.3390/s20236878_

Round 1

Reviewer 1 Report

The manuscript entitled “Breath acetone measurement based prediction of exercise induced energy and substrate expenditure” by Jeon et. al. is a nice piece of scientific literature that describes the questionable trend between breath acetone and other metabolites during fatty acid metabolism. As a whole this manuscript should be considered for publication. Below are some minor, yet important revisions that this reviewer feels should be addressed.

1) The participants were asked to refrain from smoking and drinking prior to taking their measurements; however, nowhere was it indicated if these individuals had a history of smoking. With the potential of compounds being absorbed in the tar of the lungs of smokers this information may be pertinent. 

2) Were the participants asked to have a food journal during this study to ensure that they were consuming less than 80 g of carbohydrates per day? If so this should be indicated.

3) The authors utilized a SnO2 NR-based sensor (page 3 lines 100 to 110) and described briefly how it was fabricated. A reference should be included here. Also, was this device calibrated over a set concentration range of acetone vapor, and were these results validated with a more traditional GC/MS device?

4) The conclusion is rather light on content. It should be expanded upon somewhat. 

Overall, this is a very nice piece of literature and should be published after minor revisions.

Author Response

Response to the Reviewers

Reviewer #1 (Reviewer Comments to the Author):
Reviewer’s comments

Overall comments from the reviewer #1.

The manuscript entitled “Breath acetone measurement based prediction of exercise induced energy and substrate expenditure” by Jeon et. al. is a nice piece of scientific literature that describes the questionable trend between breath acetone and other metabolites during fatty acid metabolism. As a whole this manuscript should be considered for publication. Below are some minor, yet important revisions that this reviewer feels should be addressed.

We thank the reviewer for positive and constructive comments.

The participants were asked to refrain from smoking and drinking prior to taking their measurements; however, nowhere was it indicated if these individuals had a history of smoking. With the potential of compounds being absorbed in the tar of the lungs of smokers this information may be pertinent.

Response: We thank the reviewer for this important comment. Fortunately, none of our participant was a smoker. To increase the clarity of our manuscript, we mentioned in the result the none of our participant was a smoker.

Were the participants asked to have a food journal during this study to ensure that they were consuming less than 80 g of carbohydrates per day? If so this should be indicated.

Response: We thank the reviewer for this comment. Our participants kept the food journal for two days prior to the study and this food journal was handed in during their second visit. As recommended, we indicated in our method section regarding the food journal as follows:

“As the amount of glycogen stored in the liver and muscles might influence substrate utilization, the participants were asked to consume less than 80 g of carbohydrate (CHO) per day for two days [24]. After checking their food journal to make sure that they consumed less than 80 g of CHO per day for two days prior to the second visit, the participants ran on a treadmill for one hour at a moderate intensity (70% VO2max).” (page 3. Line 87-91)

The authors utilized a SnO2 NR-based sensor (page 3 lines 100 to 110) and described briefly how it was fabricated. A reference should be included here. Also, was this device calibrated over a set concentration range of acetone vapor, and were these results validated with a more traditional GC/MS device?

Response: As recommended, we have added the following reference. In addition, we have also added the following sentences in the manuscript:

“BrAce analyzer used in our experiment was validated with GC-Mass Spectrometry (MS) and prior to each test, the device was calibrated ranged from 0.1-50 ppm of acetone vapor [31].” (Page 4. Line 128-130)

The conclusion is rather light on content. It should be expanded upon somewhat.

Response: We thank the reviewer for this important suggestion. As suggested, we have expanded  our discussion thoroughly.

Reviewer 2 Report

General comments:

I would like to thank Kim et al for their letter on “Breath acetone measurement-based prediction of 2 exercise-induced energy and substrate expenditure”.  It is a well-written and presented piece of work regarding the validation of a portable device containing an SnO2 nanorod sensor.  I thought the sensor and column use, and paragraph describing the device, was very interesting.

However, I feel the letter falls in between being a significant contribution to the sensors application discipline and the physiological use of such devices.  Similar work has been done, but is not referenced or referred too, it may have a slightly different exercise protocol (e.g. intensity/duration), and may not have focused exclusively on acetone, but acetone exercise profiles have been published.  In addition, devices have been created which measure acetone breath acetone levels, albeit using different techniques.  The authors need to justify why their approach and why such use is an improvement on current methods for achieve this goal.

Minor comments:

Abstract:  well written: change “aimed”, did you or did you not achieving this; “different time-points”, please specific.

Introduction:

lines 48-54: please revisit this paragraph for use of language.  Also, review carts and other methods of measuring.

Lines 55-60: please include, and critique other studies on breath acetone

Methods:

Describe and document the ethics approval.

2.2. describe or reference your section on VO2Max protocol.

How did you ensure compliance to diet?R

Remove BMI calculation, simple state as kg/m2

Describe BIA protocol/validation/accuracy.

There is no description of how the cart measurements were taken, validation, accuracy, nor how the cart produced the result of EE FaxOx and CarbOx – what assumptions are you making?  Hint: protein.

Describe how the blood samples were taken.

Review your statistics in view that you cannot assume, nor determine normality.

Validity of conducting regressions with n=6, using repeated measurements, i.e. independence.

Results/Discussion: need considering in light of above comments.

Author Response

Reviewer #2 (Reviewer Comments to the Author):
Reviewer’s comments

Overall comments from the reviewer #2.

Overall, this is a very nice piece of literature and should be published after minor revisions. However, I feel the letter falls in between being a significant contribution to the sensors application discipline and the physiological use of such devices. Similar work has been done, but is not referenced or referred too, it may have a slightly different exercise protocol (e.g. intensity/duration), and may not have focused exclusively on acetone, but acetone exercise profiles have been published. In addition, devices have been created which measure acetone breath acetone levels, albeit using different techniques. The authors need to justify why their approach and why such use is an improvement on current methods for achieve this goal.

First of all, we thank the reviewer for his/her time to provide these constructive feedbacks, which helped us to improve our manuscript. As mentioned by the reviewer, there have been few similar studies which measured breath acetone along with other markers such as isoprene during prolonged aerobic exercise. As recommended by the reviewer, we referenced those articles and further discuss findings of those articles in our discussion. We have also added justification as to why our study is unique in protocol and device in our discussion.

well written: change “aimed”, did you or did you not achieving this;

Response: We thank the reviewer’s comments on the abstract. With regard to the first point, we had two aims: 1) to validate our device, and to verify if the breath acetone levels measured by our device could be used to predict substrate utilization during exercise. We believed that we achieved our first aim, to validate this newly developed BrAce analyzer against blood ketone levels. However, we cannot say with confidence that we have verified whether BrAce could predict substrate use. Therefore, we have changed the second aim of our study from ‘verify’ to ‘explore’. Now, the purpose of our study is as follows:

“The purpose of our study was to validate a newly developed breath acetone (BrAce) analyzer, and to explore if BrAce could predict aerobic exercise-related substrate use.” (page 1. Line 16-17)

“different time-points”, please specific.

Response: We thank the reviewer for this important point. However, we could not be more specific about those specific time points due to word limit (less than 200 words)

lines 48-54: please revisit this paragraph for use of language. Also, review carts and other methods of measuring.

Response: As recommended, we pay careful attention on these sentences and improved as follows:

“However, this device can only be used in the laboratory and also very expensive. Recently, wearable devices provide information on energy expenditure based on heart rates and accelerometry data [11, 12], but are not able to provide substrate utilization during exercise. In this regard, breath acetone became a good candidate to be used to estimate the substrate utilization during exercise or PA [13, 14]. (page 2, line 49-52)

Lines 55-60: please include, and critique other studies on breath acetone.

Response: As recommended, we have revised our manuscript.  

Due to the volatile nature of acetone, it fills the lungs and is exhaled with the breath [16,17]. A few studies have reported a moderate association between BrAce and blood ketone levels [18] either at rest [19]and during ketogenic diet [20]. There was one study which examined breath acetone as well as BOHB in response to 90 min interval exercise on bicycle. Although Guntner et al. [21] reported that increase in breath acetone levels may have reflected increase in BOHB, which may suggest increased fat oxidation, they did not directly measure fat oxidation using a metabolic cart, nor measured free fatty acid in the blood.  Furthermore, they only collected blood three times throughout the course of study including at baseline, immediately post exercise and three hours after exercise cessation. Previous studies only examined correlation between BrAce and blood BOHB, yet, did not examined whether BrAce would associate with metabolic cart measured fat utilization or plasma free fatty acid concentration.  Therefore, it is necessary to study whether BrAce would be associate with other blood ketone such as ACAC or total ketone, and more importantly with blood free fatty acid and metabolic cart accessed fat utilization. 

Recently, we developed a portable-size breath analyzer using a SnO2 nanorod (NR)-based sensor with a miniaturized GC column, highly sensitive (detection limit: 0.1ppm), ideal for human breath acetone analysis [22]. Therefore, our primary purpose was to validate our newly developed BrAce analyzer at rest, during one hour of moderate-intensity exercise, and during three hours of recovery, whether BrAce would associate with blood ketones including ACAC and BOHB. Then, we further explored if BrAce could be a good indicator of substrate utilization, assessed by a metabolic cart as well as plasma free fatty acid before, during, and after aerobic exercise.” (page 2. Line 55-74)

Describe and document the ethics approval.

Response: We acknowledge that our explanation was not sufficient enough about ethics approval. As commented by the reviewer, we revised our manuscript to make it clear as follows:

“All participants were fully explained the experimental procedure by the experimenter and gave written informed consent prior to the test. This study was approved by the Institutional Review Board of Yonsei University (Yonsei IRB no. 7001988-201910-HR-674-05).” (page 2~3. Line 80-83)

2.2. describe or reference your section on VO2max protocol.

Response: We thank the reviewer for these important comments. In our study, we conducted cardiopulmonary exercise test (CPET) to measure participants’ maximal oxygen consumption and we used the value in order to set the intensity of 1 hour running exercise. However, we agree to the comment from the reviewer. Therefore, we added more detailed description on the CPET test in the methods as follows:

“The cardiopulmonary exercise test was performed on a treadmill using a computerized cardiac stress testing system (Cardiac Science, Q-stress TM65, Waukesha, WI, USA) to measure VO2max which would be used as a reference of exercise intensity. The participants wore a non-breathing facemask (Hans Rudolph, Rudolph series 7910, Kansas, MO, USA) during the test. The volume of carbon dioxide in exhaled air was continuously analyzed for single-breath measurement using a computerized metabolic measurement system (ParvoMedics, TrueOne 2400, UT, USA). A well-trained investigator followed the Bruce protocol. VO2 was considered maximal if any two of the following three criteria were met: 1) RER of > 1.15; 2) heart rate of > 85% of age-predicted maximal heart rate; and 3) perceived exertion rate of 17 on the Borg scale. Measured VO2max was used to establish the relative exercise intensity according to the participants.” (page3. Line 105-114)

How did you ensure compliance to diet?

Response: We thank the reviewer for insightful question and comments. We realized that the current form of our manuscript lack explanation on this important issue. All our participants were asked to keep the dietary intake record for three days including 2 days of low carb intake days (less than 80 g of CHO per day). Therefore, we have added more detailed description on how we ensured the amount of CHO consumption less than 80g in the methods as follows:

“As the amount of glycogen stored in the liver and muscles might influence substrate utilization, the participants were asked to consume less than 80g of carbohydrate (CHO) per day for two days [24]. After checking their food journal to make sure that they consumed less than 80g of CHO per day for two days prior to the second visit, the participants ran on a treadmill for one hour at a moderate intensity (70 % VO2max).” (page 3. Line 87-91)

Remove BMI calculation, simple state as kg/m2

Response: As recommended, we remove the calculation and simplified the unit of BMI in our manuscript.

“The body mass index (BMI) was described as kg/m2.” (page 3. Line 100-101)

Describe BIA protocol/validation/accuracy.

Response: We agree with points raised by the reviewer and have revised our manuscript by adding the previous studies as references which validated the accuracy of Inbody 720.

  1. Gibson, A. L., Holmes, J. C., Desautels, R. L., Edmonds, L. B., & Nuudi, L. (2008). Ability of new octapolar bioimpedance spectroscopy analyzers to predict 4-component–model percentage body fat in Hispanic, black, and white adults. The American journal of clinical nutrition, 87(2), 332-338.
  2. Ling, C. H., de Craen, A. J., Slagboom, P. E., Gunn, D. A., Stokkel, M. P., Westendorp, R. G., & Maier, A. B. (2011). Accuracy of direct segmental multi-frequency bioimpedance analysis in the assessment of total body and segmental body composition in middle-aged adult population. Clinical nutrition, 30(5), 610-615.
  3. McLester, C. N., Nickerson, B. S., Kliszczewicz, B. M., & McLester, J. R. (2018). Reliability and agreement of various InBody body composition analyzers as compared to dual-energy X-ray absorptiometry in healthy men and women. Journal of Clinical Densitometry

There is no description of how the cart measurements were taken, validation, accuracy, nor how the cart produced the result of EE FatOx and CarbOx – what assumptions are you making?  Hint: protein.

First and foremost, we thank the reviewer for the insightful comment. We realized that the current form of our manuscript lack explanation on this important issue. Calculating fat and CHO oxidation via indirect calorimetry have been practiced for many years in physiological studies and we might have overlooked in providing the detailed information. Therefore, we have added more detailed description on energy expenditure calculation using VO2 and RER with regard to fat and CHO oxidation in the methods with proper reference as follows:

“Energy expenditures from fat and CHO oxidation were calculated according to the non-protein respiratory quotient based on the VO2 and RER obtained from the metabolic cart [25].” (page 3. Line 93-94)

Describe how the blood samples were taken.

As recommended, we have now provided more detailed information regarding blood sample methods as follows:

“Blood samples were taken six times; at baseline, immediately after 1-hour exercise, 30 min, 60 min, 120min, and 180 min after exercise. Blood samples (10 ml each time) were collected by standard vein puncture into the plain tube. Then blood samples were centrifuged at 3000 rpm for 10 min. The serum    was separated into a separate tube and kept in −80 ◦C freezer until further analysis of the following variables: glucose, BOHB, ACAC, total ketone (BOHB+ACAC), and free fatty acid (FFA).” (page 4. Line 134-138)

Review your statistics in view that you cannot assume, nor determine normality. Validity of conducting regressions with n=6, using repeated measurements, i.e. independence.

We agree with the reviewer that with N of 6, it is not ideal to use parametric statistics since testing for normality of distribution with data from 6 participants. Therefore, we have changed the purpose of our study from ‘verify’ to ‘explore if BrAce could be an indicator of substrate utilization).

Although our study only explored whether BrAce would be an indicator of substrate utilization with 6 participants, previous studies which claimed that BrAce was a good measure for fat oxidation only used blood BOHB levels as an indicator of fat oxidation. In our study, we not only measured BOHB as well as ACAC, then further, we used indirect calorimetry (almost gold standard second to metabolic chamber) and blood free fatty acid as indicators of fat oxidation. Previous study only reported blood ketone level at baseline, after 90 min of interval exercise, and at 3 hours of recovery, which they claimed that BrAce reflect fat burn rate. Our study measured BrAce and blood ketones at 6 different time points as well as continuous measurements of substrate utilization using a metabolic cart. Therefore, we believe our data is worth reporting although there are only 6 participants. Yet, we agree with the reviewer that 6 participants are not big enough to properly validate using parametric statistics and that is why we used the word ‘explore’ rather than ‘verify’. Furthermore, we also extensively revised our manuscripts in the discussion including limitation of our study.

Results/Discussion: need considering in light of above comments.

Thanks, we have revised our results and discussion according to the comments above.

Reviewer 3 Report

This study developed a breath acetone (BrAce) analyzer using a SnO2 nanorod (NR)-based sensor to verify its ability on the prediction of the concentration of aerobic exercise-related substrate. Six men were involved in the experiment to get the data for the validation of the use of BrAce analyzer. The main problem is this validation study might not be suitable to publish on Sensors since there were no related descriptions on the sensor and the analyzer. Furthermore, the sample size is too small to draw conclusions.

Other comments:

  • The authors provided the structure of the analyzer in Fig 1, but no calibration data was provided for the evaluation of the performance of this analyzer, especially on the sensitivity, selectivity, and reproducibility.
  • The details of the analyzer were not provided, such as the operating temperature of the separation column, the procedure of the working valve and pump, as well as the SnO2 nanorod (NR)-based sensor. Without the information, the readers cannot reproduce the work.
  • The statistical power of this designed experiment is not clear. The sample size calculation was missing. The interference of race and gender was not considered in the experiment design, thus a limitation should be addressed for the prediction model.
  • Page 1. Lines 17: How did you get the value of 70% of maximum oxygen consumption?
  • Page 2. Lines 64: Portable breathalyzers require more accurate equipment, such as GC-MS, to verify the accuracy of the obtained data.
  • Any ethical approval or written informed consent from the participants was obtained since the blood samples were needed in the study.

Author Response

Reviewer #3 (Reviewer Comments to the Author):
Reviewer’s comments

Overall comments from the reviewer #3.

This study developed a breath acetone (BrAce) analyzer using a SnO2 nanorod (NR)-based sensor to verify its ability on the prediction of the concentration of aerobic exercise-related substrate. Six men were involved in the experiment to get the data for the validation of the use of BrAce analyzer. The main problem is this validation study might not be suitable to publish on Sensors since there were no related descriptions on the sensor and the analyzer. Furthermore, the sample size is too small to draw conclusions.

Most of all, we thank the reviewer for the meaningful questions and comments. Our group has already developed and tested its validity of the device previously, which we have published (Jung et al. 2018).  All detailed information regarding the description of the sensors have been previously reported and therefore, we did not include related descriptions on the sensor in this submitted manuscript. Furthermore, the special issue which we have submitted is “Sensors for Exercise and Sport Activities: From Health Promotion to Sports Performance” and also our manuscript was a form of ‘Letter’ instead of ‘Original article’.

Regarding the small sample size, we agree with the reviewer that our sample size is too small for validation purpose. Therefore, we have changed the purpose of our study from ‘verify’ to ‘explore if BrAce could be an indicator of substrate utilization). Although our study only explored whether BrAce would be an indicator of substrate utilization with 6 participants, previous studies which claimed that BrAce was a good measure for fat oxidation only used blood BOHB levels as an indicator of fat oxidation. In our study, we not only measured BOHB as well as ACAC, we also used indirect calorimetry (almost gold standard second to metabolic chamber) and blood free fatty acid as indicators of fat oxidation. Previous study only reported blood ketone level at baseline, after 90 min of interval exercise, and at 3 hours of recovery, which they claimed that BrAce reflect fat burn rate. Our study measured BrAce and blood ketones at 6 different time points as well as continuous measurements of substrate utilization using a metabolic cart. Therefore, we believe our data is worth reporting although there are only 6 participants. Yet, we agree with the reviewer that 6 participants are not big enough to properly validate the hypothesis of our study using parametric statistics and that is why we used the word ‘explore’ rather than ‘verify’.  Lastly, we also extensively revised our manuscripts in the discussion including limitation of our study.

The authors provided the structure of the analyzer in Fig 1, but no calibration data was provided for the evaluation of the performance of this analyzer, especially on the sensitivity, selectivity, and reproducibility.

Response: We agree with the reviewer that we didn’t supply the details of the device including the sensitivity, selectivity, and reproducibility in the manuscript. However, we have now provided the content by adding the patent of our device as a reference.

“BrAce analyzer used in our experiment was validated with GC-Mass Spectrometry (MS) and prior to each test, the device was calibrated ranged from 0.1-50 ppm of acetone vapor [31].” (page 4. Line 128-130)

The details of the analyzer were not provided, such as the operating temperature of the separation column, the procedure of the working valve and pump, as well as the SnO2 nanorod (NR)-based sensor. Without the information, the readers cannot reproduce the work.

Response: We agree with the reviewer that the current manuscript does not provide this information. Therefore, we have now revised our methods as well as referred previously published manuscript which included this information.

This is what we have described in the previously published paper (Jung et al. 2018, Sensors and Actuators: B Chemical) and patent.

Breath acetone analyzer

The breath acetone analyzer comprised a sampling loop, a packed column, three solenoid valves, a mini-sized pump, and a sensor based on ZnO QDs. Fig. 1 shows a schematic of the analyzer and an optical image of the real breath analyzer. The dimensions of the breath acetone analyzer were 8 × 13 × 16 cm3 . First, the sampling loop was filled with 1 ml exhaled breath without pre-concentration. Subsequently, acetone was separated from the exhaled breath by the packed column within 100 s at a carrier gas flow rate of 20 sccm. The operating temperature of the column was maintained at 30 °C. The separated acetone was detected by the ZnO QD-based sensor. The length and inner diameter of the packing column (Isenlab Inc.) were 20 cm and 0.15 cm, respectively. The ZnO QDs were synthesized by a wet chemical method. The ZnO precursor solution was prepared using Zn acetate (Zn(O2CCH3)2, Alfa Aesar). In brief, 1.975 g Zn acetate was dissolved in 90 ml N,N-dimethylformamide ((CH3)2NC(O)H), Duksan). The ZnO precursor solution was injected into a methanolic solution of tetramethylammonium hydroxide (N(CH3) 4+(OH)− (methanol:tetramethylammonium hydroxide = 1:8) using a syringe pump for 1 h at 30 °C. The synthesized ZnO QDs were rinsed with acetone and dispersed in methanol. The ZnO QDbased sensor was prepared by dispersing the ZnO QD solution on an Al2O3 substrate, followed by heat treatment at 350 °C for 30 min to remove the residue. The microstructure of the ZnO QDs was investigated by scanning electron microscopy (SEM, JEOL-7001 F) and tunneling electron microscopy (TEM with energy-dispersive X-ray spectroscopy (EDX), JEM-F2000), which were also used for the compositional analysis of the ZnO QDs.

Sensor Characterization

To fabricate acetone sensors, the ZnO QDs were dispersed on an alumina substrate (0.5 × 0.25 mm2 ) supplied with interdigitated Pt electrodes and heating elements. The devices were dried at 90 °C and heat-treated at 350 °C for 30 min to remove the residual organic compounds. The ZnO QD sensors were tested at various temperatures (394, 417, 430, and 446 °C) to optimize the operation temperature of the breath acetone analyzer system. The breath acetone analyzer was used to detect air-balanced acetone. The concentration of air-balanced acetone was in the range 0.1–50 ppm at the optimal temperature of 430 °C. Acetone was directly injected into the breath acetone analyzer, without any pre-concentration, for 10 s. The sampling volume of the acetone was limited to 1 ml. Ambient air was used as the carrier gas at a flow rate of 30 sccm. The resistance of the ZnO QD sensor was converted to a sensor signal (log(R)) by the breath acetone analyzer. The response of the ZnO QD sensor was defined as Δ(log(R)): Δ(log(R)) = log(R)max − log(R)min (1) where log(R)max is the maximum resistance before exposure of the acetone and log(R)min is the minimum resistance at exposure. These values are related to the resistance of the ZnO QD sensor”.

And, we have revised our manuscript as follows:

“The BrAce analyzer was composed of a sampling loop, a packed column, three solenoid valves, a mini-sized pump, and a SnO2 NR-based sensor. The detailed explanation of the sensors, characterization and procedure of the working valve and pump have been previously described [22].” (page 3. Line 118-120)

The statistical power of this designed experiment is not clear. The sample size calculation was missing. The interference of race and gender was not considered in the experiment design, thus a limitation should be addressed for the prediction model.

Response: We fully agree with the reviewer that our manuscript is missing sample size calculation and participants included in our study are single race and gender. As recommended by the reviewer, we will include these as limitation of our study.

Page 1. Lines 17: How did you get the value of 70% of maximum oxygen consumption?

Response: We thank the reviewer for this question.  First of all, we have tested participants’ maximal oxygen consumption (VO2 max), then used 70% of their VO2 max as a target intensity. For example, if participants’ VO2 max was 50 ml/min/kg body weight, then 70% of their VO2 max would be 35 ml/min/kg body weight. When participants ran on a treadmill for 1 hour, we continually monitor his VO2 levels, and when VO2 levels fell below 35 ml/min/kg body weight, then we would increase the speed of the treadmill. Or, when VO2 levels went over 35 ml/min/kg body weight, then we would decrease the speed of the treadmill. The moderate intensity exercise based on VO2 max would be between 60-80 % VO2 max and above 80% VO2 max would be considered as vigorous intensity. If we chose intensity of exercise too high, then, participants are not able to run for 1 hour.  Therefore, based on our pre-testing (pilot), we determined 70% of the maximal oxygen consumption as the intensity of our experiment.

Page 2. Lines 64: Portable breathalyzers require more accurate equipment, such as GC-MS, to verify the accuracy of the obtained data.

Response: The verification of the accuracy of the device has previously determined and published.

Any ethical approval or written informed consent from the participants was obtained since the blood samples were needed in the study.

Response: We thank the reviewer for this comment. We have obtained institutional ethical approval and also acquired written informed consent form from our participants. Accordingly, we revised our manuscript to make it clear as follows (page 2).

“All participants were fully explained the experimental procedure by the experimenter and gave written informed consent prior to the test. This study was approved by the Institutional Review Board of Yonsei University (Yonsei IRB no. 7001988-201910-HR-674-05).” (page 2~3. Line 80-83)

Round 2

Reviewer 2 Report

All points addressed; accept for publication.

Reviewer 3 Report

The authors answered all of my questions.